# Personalized Immunity: Neoantigen-Based Vaccines Revolutionizing Hepatocellular Carcinoma Treatment

**DOI:** 10.3390/cancers17030376

**Published:** 2025-01-23

**Authors:** Ioanna Aggeletopoulou, Spyridon Pantzios, Christos Triantos

**Affiliations:** 1Division of Gastroenterology, Department of Internal Medicine, University Hospital of Patras, 26504 Patras, Greece; chtriantos@upatras.gr; 2Hepatogastroenterology Unit, Academic Department of Internal Medicine, General Oncology Hospital of Kifissia “Agioi Anargyroi”, National and Kapodistrian University of Athens, 14564 Athens, Greece

**Keywords:** neoantigen, neoantigen-based vaccine, HCC, adjuvant treatment, personalized therapy

## Abstract

Liver cancer is a complex and difficult disease to treat, often detected at advanced stages and resistant to standard therapies. This review focuses on a promising approach using personalized vaccines targeting neoantigens. By activating the immune system to specifically attack these tumor-specific targets, neoantigen vaccines offer a highly targeted and effective strategy. This study examines how these vaccines are developed, their integration with other therapies like immune checkpoint inhibitors, and their potential to improve patient outcomes. Early studies show encouraging results in enhancing the immune response and controlling the disease. Despite challenges such as tumor variability and the complexity of vaccine production, this approach represents a major step toward precision medicine in liver cancer. These findings highlight the need of further advancements in personalized therapies, providing hope for better survival rates and long-term control of this challenging disease.

## 1. Introduction

Liver cancer is the sixth most prevalent malignancy and the second leading cause of cancer-related mortality globally, presenting a critical public health challenge [1,2,3]. Hepatocellular carcinoma (HCC) accounts for over 90% of primary liver cancer cases, making it the most common form of the disease [4]. Despite advancements in treatment strategies, the prognosis for HCC remains poor, with a five-year survival rate of less than 20%, and a recurrence rate of up to 88% [5]. This highlights the urgent need for more effective therapeutic approaches.

The etiology of HCC is multifaceted, with major risk factors including chronic hepatitis B and C infections, alcoholic liver disease, and metabolic dysfunction-associated steatotic liver disease (MASLD) [6]. Additional contributing factors such as tobacco use, dietary aflatoxins, and rare monogenic disorders make its pathogenesis more complicated [7]. Nearly 90% of HCC cases occur in the setting of chronic liver disease, with cirrhosis serving as the predominant risk factor for its onset [4,8,9]. While prompt detection allows for potentially curative interventions such as surgical resection, local ablation, or liver transplantation, these alternatives are limited to individuals with preserved liver function [4,8]. For intermediate stages, transarterial chemoembolization (TACE) is suggested, whereas systemic and immune therapies are the primary options for advanced HCC [4,10,11]. However, even in early stages, recurrence rates can reach up to 70% within five years, and the survival outcomes for advanced HCC remain poor [12].

Immunotherapy has made a significant advancement in cancer treatment by activating the immune system to identify and destroy tumor cells [13]. In HCC, immune checkpoint inhibitors (ICIs), which block proteins including PD-1, PD-L1, and CTLA-4, have emerged as a cornerstone of systemic therapy, but their efficacy is limited by low T cell infiltration [14] and the tumor’s modest mutational burden [15]. ICIs as monotherapy show response rates of only 15–20% in HCC [16,17], emphasizing the need for strategies to enhance tumor immunogenicity.

The landscape of immunotherapy has been revolutionized with the advent of personalized cancer vaccines. These vaccines have been classified into two core categories: preventive and therapeutic [18]. Preventive vaccines are engineered to impede cancer development by targeting oncogenic pathogens such as human papillomavirus (HPV) and hepatitis B virus (HBV), effectively reducing cancer incidence [18,19]. On the other hand, therapeutic vaccines are designed to combat existing cancers by activating the immune system to selectively identify and destroy malignant cells [18,20,21]. In HCC, vaccine-based therapies have emerged as a groundbreaking treatment option, offering a novel approach to control and potentially eradicate the tumor. These therapies aim to establish durable tumor suppression and prolonged remission by inducing the immune system to systematically recognize and attack tumor cells [20,21].

The aim of the current review is to explore and highlight the therapeutic potential of neoantigen-based vaccines in the management of HCC. Specifically, it summarizes the process of neoantigen identification and their role in enhancing immunogenicity against HCC. It evaluates the current evidence on various vaccine platforms and discusses the integration of neoantigen vaccines with ICIs to overcome the immunosuppressive tumor microenvironment in HCC. Additionally, the review examines preclinical and clinical data to assess the efficacy and safety of these personalized therapies, while identifying challenges in translating neoantigen vaccines into clinical practice.

## 2. Tumor Neoantigens

The accumulation of somatic mutations within the tumor genome serves as a key indicator of responsiveness to ICI therapies. These mutations, referred as tumor mutational burden (TMB), primarily result from unrepaired DNA damage, leading to the release of aberrant proteins displayed on the surface of tumor cells [22]. Among these aberrant proteins are tumor-specific antigens (TSAs), which include neoantigens generated directly by somatic mutations, as well as TSAs derived from other tumor-restricted processes, such as viral oncogenes. Neoantigens are presented on the cell surface by major histocompatibility complex (MHC) molecules, forming the peptide-MHC (pMHC) complexes that can be recognized by T cell receptors (TCRs), which alongside the expression of co-stimulating molecules and cytokine release, can induce an immune response against cancer [23,24,25,26]. However, tumor cells have developed mechanisms to evade immune detection [27]. Immunotherapies aim to overcome these immune escape mechanisms. These include ICIs to restore CD8+ T-cell function, tumor vaccines to enhance antigen presentation, adoptive transfer of tumor-infiltrating lymphocytes (TILs) or T cells with engineered TCRs, chimeric antigen receptor (CAR)-modified T cells, bispecific antibodies (bsAbs), and strategies to modulate the tumor immune microenvironment [28,29,30].

### 2.1. Mechanisms of Neoantigen Generation

Tumor-associated antigens (TAAs) have been explored as targets for immunotherapies so far; TAAs are normal proteins abnormally expressed in cancer tissues or at specific stages of differentiation, with minimal expression in normal tissues [31]. However, since TAAs are non-mutated self-antigens, the immune system’s tolerance to them often results in weak T cell responses in the clinical setting [32]. In contrast, neoantigens are generated through various mechanisms, each contributing uniquely to their diversity and immunogenicity (Table 1).

One major pathway involves genetic alterations, such as somatic mutations, which result in the production of single nucleotide variants (SNVs). These non-synonymous mutations result in amino acid substitutions. SNVs have gained great interest following the discovery that TMB of non-synonymous mutations significantly correlates with the response to checkpoint inhibitors [33]. The increased immunogenicity of SNV-derived epitopes is attributed to the introduction of a new amino acid, which either enhances the interaction with the TCR or creates a novel epitope with improved binding and presentation capacity via MHC molecules [34]. SNV-derived neoantigens in HCC exhibit increased immunogenicity, highlighting their potential for effective vaccine strategies in HCC [35]. Specifically, a study identified a median of 30 SNV-derived mutations per patient, with 66% of tested HLA-A*02.01-restricted peptides inducing robust CD8+ T cell responses in murine models [35]. Peptides like KYV and ALL showed strong T cell activation, selectively targeting mutated sequences over wild-type [35]. Another mechanism involves insertions or deletions in somatic cells. Higher prevalence of mutations caused by insertions or deletions (indels) often cause stronger responses to checkpoint inhibitors in tumors [36]. Notably, studies have revealed that nearly 10% of MHC-I-presented ligands are peptides originating from indels [37]. Chromosomal translocations also contribute by generating neoepitopes with mutations at the breakpoint; however, the limited natural processing and presentation of these ligands may explain the low response rates observed with vaccines targeting such neoantigens [38]. In addition to genetic changes, transcriptional mechanisms, including RNA alternative splicing, significantly expand the immunopeptidome. Data have shown that epitopes generated through alternative splicing have made a significant contribution to this repertoire [39]. Furthermore, translational processes, including initiation at non-AUG start codons and ribosomal frameshifting, provide another source of neoantigen diversity [23,40]. Lastly, post-translational modifications, encompassing antigen processing and proteasomal peptide cleavage, play a substantial role in shaping the immunogenic landscape [23,40].

### 2.2. Neoantigen Presentation, Immune Activation, and Challenges Related to Immunogenicity

Neoantigens, released by damaged or dying tumor cells, are recognized by antigen-presenting cells (APCs). In the lymph nodes, APCs cross-present tumor-associated antigens via MHC class I molecules, initiating the activation of cytotoxic CD8+ T cells through a process called cross-priming [41,42]. Activated CD8+ T cells then migrate to the site of the tumor, where they perform their cytotoxic functions in an antigen-specific manner, potentially amplifying the immune response through epitope spreading [41,42]. Meanwhile, CD4+ T cells activated by tumor antigens presented on MHC class II molecules enhance the activity of CD8+ T cells, promote MHC class I expression on tumor cells, stimulate cytotoxicity of myeloid cells, and finally contribute to the targeted destruction of tumor cells [41,42].

Mutations can also generate non-immunogenic neoantigens. Proper peptide cleavage and sufficient MHC binding affinity are essential for neoantigen presentation and subsequent recognition by TCRs. Linnemann et al., demonstrated that only 0.5% of mutated peptides are recognized by TILs [43]. Additionally, the neoantigen repertoire within the tumor is shaped by immune interactions through a process known as cancer immunoediting. This process involves immune selection, where T cells eliminate cells expressing highly immunogenic neoantigens while allowing those with lower immunogenicity to survive [44]. Furthermore, other mechanisms contribute to immune evasion, including the upregulation of immune checkpoint molecules such as PD-L1, secretion of immunosuppressive cytokines like TGF-β and IL-10, recruitment of regulatory T cells (Tregs) and myeloid-derived suppressor cells (MDSCs), and downregulation of MHC molecules, which collectively impair effective neoantigen presentation and T cell recognition [45,46].

Lastly, clonality plays a critical role in determining neoantigen properties. Clonal neoantigens develop during the early stages of tumorigenesis and are found in the majority of tumor cells, whereas subclonal neoantigens develop later and in fewer cells. Evidence has shown that patients with a higher proportion of clonal neoantigens exhibit improved survival rates and intensified response to ICIs compared to those with predominantly subclonal neoantigens [47].

Unlike TAAs, neoantigens are expressed in specific tumors and are entirely absent from healthy tissues, making them optimal candidates for personalized cancer treatments [48]. Additionally, T cells specific to neoantigens can bypass the thymic negative selection, allowing for robust and tumor-specific immune responses [23,49]. Moreover, the combination of neoantigen-specific T cell responses with immunotherapies may provide durable immunological memory, offering the potential for long-term protection against cancer recurrence [50]. Several malignancies, such as non-small cell lung cancer [51], melanoma [52], pancreatic cancer [53], and urothelial carcinoma [54], have demonstrated enhanced responses to ICIs when associated with higher TMB and/or an increased neoantigen load. However, in HCC, TMB is considered intermediate to low (<5 somatic mutations/megabase), with approximately 60 non-synonymous mutations within the coding regions of the exome, which limits its reliability as a predictive biomarker for response to ICIs [55].

## 3. Neoantigen as Targets for Hepatocellular Carcinoma

### 3.1. Genetic and Molecular Landscape of Hepatocellular Carcinoma

Advancements in genomic and molecular analyses have provided valuable insights into the complex genetic landscape of HCC, highlighting key mutations, pathways, and prognostic factors associated with tumor progression and therapeutic responses. Cancer patients can exhibit hundreds of nonsynonymous somatic mutations, leading to an average of approximately 150 potential neoantigens per individual [56]. In HCC, a study using RNA-seq analysis and two prediction servers revealed the identification of nine true predicted neoantigens [57], whereas another study demonstrated the existence of 15 neoantigens per HCC patient [58]. Despite these predictions, their actual immunogenicity has not been verified, as most of the identified neoantigens have been generated through computational models without further experimental validation. Proteomic analyses by mass spectrometry, which aim to detect neoantigen peptides bound to HLA molecules in HCC tissues, have been unable to consistently confirm their presence [59]. This limitation may arise from the technical challenges associated with these methods, including the overwhelming abundance of self-antigens in tumor cells, which can obscure the detection of tumor-specific epitopes.

Experimental confirmation efforts have predominantly focused on evaluating T cell recognition. Notably, recent studies have demonstrated that TILs are capable of recognizing neoantigens in tumors originating from the same patient [60]. Additionally, neoantigen-specific T cells have been successfully retrieved from both tumor tissues and peripheral blood of HCC patients, highlighting the effectiveness of current neoantigen identification methods in recognizing these targets [61]. These findings underline the potential of neoantigen-focused immunotherapies while also emphasizing the need for more robust experimental validation.

Exome sequencing analysis in HCC patients revealed 161 putative driver genes associated with 11 modified pathways, including mutations on the TERT promoter (60%), which activate telomerase expression, as well as alterations in WNT/β-Catenin (54%), PI3K/AKT/mTOR (51%), TP53/cell cycle (49%), MAP kinase (43%), hepatic differentiation (34%), epigenetic regulation (32%) chromatin remodeling (28%), oxidative stress (12%), IL-6/JAK/STAT (9%), and TGF-β (5%) [62]. Risk factor-specific mutational signatures were identified, with CTNNB1 mutations linked to alcohol use, TP53 mutations associated with HBV infection, and AXIN1 alterations connected to other risk factors [62]. TERT promoter mutations were observed as early events in tumor development, while amplifications in FGF3/4/19 or CCND1 and alterations in TP53 and CDKN2A were characteristic of later-stage aggressive tumors [62].

These pathway abnormalities contribute to the immunosuppressive microenvironment of HCC, impacting the generation of neoantigens and the efficacy of neoantigen-based vaccines. For instance, TP53 mutations and defects in chromatin remodeling influence genomic instability [63], potentially increasing the mutational burden and resulting in the generation of novel neoantigens. Similarly, aberrations in WNT/β-Catenin signaling have been linked to reduced T cell infiltration [64], thereby limiting the immune system’s ability to recognize and target neoantigen-expressing tumor cells. Moreover, alterations in the IL-6/JAK/STAT and PI3K/AKT/mTOR pathways contribute to immune evasion by upregulating immunosuppressive cytokines and reducing T cell functionality [65], which may affect the effectiveness of neoantigen vaccines.

Whole-exome sequencing and DNA copy number analyses in 363 HCC patients accompanied by DNA methylation, RNA, miRNA, and proteomic expression in 196 HCC patients identified mutated genes and alterations that may contribute to metabolic reprogramming [66]. Through integrative molecular subtyping, the analysis revealed a p53 target gene expression signature associated with poorer prognosis [66]. Additionally, potential therapeutic targets, including immune checkpoint proteins and WNT signaling, MDM4, MET, VEGFA, MCL1, IDH1, and TERT—which are relevant to neoantigen-driven immune responses—have been highlighted [66]. The mutational signature can vary depending on the etiology of HCC and other patient-specific factors [67], which is a critical aspect when refining neoantigen identification methods and selecting candidates for vaccine development. These findings underscore the interplay between pathway alterations, neoantigen generation, and immune responses, paving the way for targeted approaches in neoantigen vaccine development for HCC.

### 3.2. Prognostic Controversies of Tumor Mutational Burden and Neoantigen Load in Hepatocellular Carcinoma

While genomic and molecular analyses have revealed crucial pathways and therapeutic targets in HCC, the prognostic significance of TMB and neoantigen load remains debatable, with studies reporting conflicting results. A retrospective study showed that higher TMB was linked to poor prognosis in a cohort of 128 HCC patients who underwent radical hepatectomy [68]. Opposite results were reported by Liu et al., who analyzed data from The Cancer Genome Atlas and suggested that higher TMB levels were associated with improved overall survival (OS) in HCC and correlated with earlier pathological stages [69]. Similarly, high neoantigen burden has shown no consistent correlation with progression-free survival in a cohort of HCC patients who did not undergo immunotherapy, further questioning its role as a reliable biomarker in this subgroup of patients [70]. These discrepancies may be attributed to differences in patient populations, such as etiology (e.g., viral hepatitis vs. metabolic dysfunction-associated steatotic liver disease), tumor stages, and underlying liver conditions. Additionally, variations in sample sizes, sequencing platforms, and computational tools for neoantigen prediction could have influenced the findings. Biological mechanisms could also play a role; for example, in early-stage HCC, higher TMB may enhance immune activation by increasing neoantigen load, leading to improved OS. Conversely, in advanced HCC, higher TMB might reflect tumor evolution and immune escape, contributing to poor prognosis. Differences in immune responses, such as robust T cell infiltration in earlier stages compared to immune exhaustion or suppression in advanced stages, may further explain these conflicting outcomes.

However, specific subsets of neoantigens appear to hold great promise as prognostic and therapeutic markers. Whole-exome sequencing, RNA sequencing, and computational bioinformatics revealed that TP53-specific neoantigens in HCC patients were associated with longer OS and enhanced anti-tumor immunity, characterized by higher immune scores, increased cytotoxic lymphocyte infiltration, and higher cytolytic activity scores [71]. In contrast, TMB and overall neoantigen load did not correlate with prognosis, suggesting that TP53 neoantigens, rather than TMB or neoantigen load, play a key role in the prognosis of HCC patients [71]. These findings may be explained by the fact that TP53-specific neoantigens are effectively presented on MHC class I molecules, which are recognized by CD8+ T cells, leading to their activation and proliferation [72,73]. This interaction enhances cytotoxic lymphocyte infiltration, contributing to robust anti-tumor immunity [72,73]. Furthermore, immune checkpoint markers such as PD-1 and CTLA-4 may influence this interaction. Inhibition of these checkpoints using ICIs restores T cell functionality and prevents exhaustion, thereby amplifying the immune response driven by TP53-specific neoantigens.

The association between TP53-derived neoantigens and longer survival raises the question of whether combining TP53-derived neoantigen vaccines with ICIs could improve therapeutic outcomes. While specific trials investigating this combination in HCC are currently limited, preclinical and early-phase clinical studies in other cancers have suggested that neoantigen vaccines can synergize with ICIs to enhance anti-tumor immunity. For example, personalized neoantigen vaccines have been shown to increase T cell activation and tumor infiltration, which may complement the immune checkpoint blockade provided by anti-PD-1/PD-L1 therapies [74]. Personalized neoantigen vaccines have been shown to enhance T cell activation and tumor infiltration, complementing immune checkpoint blockade therapies [75]. The KEYNOTE-942 trial (a phase 2b study of mRNA-4157 [V940] with pembrolizumab in resected high-risk melanoma) provided the first clinical evidence of mRNA-based cancer treatments combined with ICIs [75]. The combination therapy improved recurrence-free survival (HR 0.561; *p* = 0.053) with an 18-month recurrence-free survival rate of 79% versus 62%, and demonstrated a manageable safety profile [75]. These findings highlight the potential of integrating mRNA vaccines targeting neoantigens, such as TP53, with ICIs to overcome the immunosuppressive tumor microenvironment of HCC. Future studies focusing on HCC-specific contexts are needed to evaluate the efficacy of such combinations, particularly with TP53-derived neoantigen vaccines.

High affinity neoantigens (HANs) have also emerged as critical players, showing a stronger correlation with OS compared to TMB or total neoantigen load [61]. HANs are neoantigens with high binding affinity to MHC-I molecules (IC50 < 50 nM), making them more likely to elicit strong tumor-specific immune responses [61]. Their high binding affinity ensures efficient presentation by APCs and robust recognition by CD8+ T cells, leading to enhanced activation, proliferation, and cytolytic activity of TILs. This distinguishes HANs from other neoantigens, which may have lower binding affinity and fail to elicit similarly strong immune responses [61]. Their prediction involves analyzing somatic mutations from whole-exome sequencing (WES) data using tools like NetMHCpan, which filter peptides based on binding affinity [76]. This process ensures the identification of highly immunogenic neoantigens crucial for understanding tumor-specific immunity. Liu et al., demonstrated that HANs were associated with enhanced anti-tumor activity through the activation of CD39 + CD8+ TILs [61]. Patients with HANs also showed improved responses to anti-PD-1 therapy, suggesting the potential of HANs as biomarkers for personalized immunotherapy in HCC [61]. Leveraging HANs for targeted immunotherapy strategies may involve prioritizing their inclusion in neoantigen vaccine designs and integrating them with ICIs to amplify tumor-specific T cell responses and overcome immune evasion.

Although more studies are needed especially in patients receiving immunotherapy with ICIs (anti-PDL1, anti-PD1, anti-CTLA4) in HCC, the latter findings highlight the intricate role of neoantigens in HCC, suggesting that while TMB and overall neoantigen load may have limited prognostic utility, high-affinity neoantigens and TP53-specific neoantigens could serve as valuable biomarkers and therapeutic targets. This highlights the need for more precise immunotherapy strategies, utilizing these specific neoantigen profiles to tailor personalized treatments for HCC.

## 4. Neoantigen Vaccines in Hepatocellular Carcinoma

Beyond the close relationship with the treatment outcome, neoantigen determination constitutes a promising immunotherapeutic approach aimed at improving the effectiveness of ICIs in HCC patients, particularly those with low tumor-infiltrating lymphocytes. Resident memory cytotoxic CD8+ T cells, which are essential for controlling tumors, can be replenished by circulating CD8+ T cells [77,78]. Essentially, when these tumor-fighting CD8+ T cells are depleted or not present in the tumor environment, circulating CD8+ T cells from the bloodstream can migrate to the tumor site and replenish the population of memory T cells, helping to maintain immune surveillance and tumor control [77,78].

However, neoantigen-MAGE-A-specific CD8+ T cells are present in about 50% of the patients, whereas neoantigen-(NY-ESO-1 and Glypican-3)-specific circulating CD8+ T cells are present in less than 15% of HCC cases [79]. To overcome this limitation, neoantigen vaccine platforms, including DNA-based, RNA-based, peptide-based, and dendritic cell (DC)-based vaccines along with adjuvants, have been developed and are currently undergoing clinical trials [80].

The development of vaccine platforms was initially explored in preclinical models using advanced genomic technologies, to identify SNVs expressed in murine tumor cell lines [50]. Mutated peptide sequences were analyzed using MHC-binding prediction algorithms [81] or identified via mass spectrometry analysis [82]. The peptides were then synthesized and assessed in immunization experiments, showing their ability to trigger neoAg-specific T cell responses and successfully slow down tumor progression after vaccination [83,84]. Figure 1 demonstrates how advanced genomic technologies are utilized to identify tumor-specific mutations, predict HLA-binding peptides, and validate their immunogenic potential through preclinical processes.

Neoantigen-based vaccines for HCC include peptide-based vaccines, which use synthetic peptides derived from tumor-specific neoantigens to induce the activation of T cell responses through the presentation on MHC molecules [85]. Plasmid DNA-based vaccines encode neoantigens for expression in host cells, enabling sustained immune activation [86,87]. However, plasmid DNA-based vaccines may pose higher risk of DNA integration into the genome of the host cell compared to RNA vaccines. In RNA-based vaccines mRNA is translated into the host cells, and the proteins produced are processed and presented on MHC molecules [88]. DC-based vaccines involve loading patient-derived DCs with neoantigens ex vivo to induce robust immune responses upon reinfusion into the patients [89]. Viral vector-based vaccines use engineered viruses to deliver neoantigen genes [90], while nanoparticle-based vaccines encapsulate neoantigens for targeted delivery to APCs [91] (Figure 2).

Various vaccination strategies including peptide-based vaccines, RNA multi-epitope vaccines, and DC-based vaccines are actively being investigated in clinical trials for HCC. In Table 2, detailed information on these strategies, including vaccine types, clinical trial phases, and their current status are reported.

From these studies, several have already reported their initial findings. A case report study derived from the clinical trial with identifier NCT03199807 demonstrated the effectiveness of combining tomotherapy with neoantigen-reactive T cells in treating advanced HCC [92]. The patient, a 75-year-old male with chronic hepatitis B, had multiple metastatic liver lesions following initial surgical and interventional treatments [92]. After receiving tomotherapy followed by neoantigen-reactive T cell therapy over four cycles, significant tumor shrinkage and stabilization were observed, along with reductions in tumor markers and increased lymphocyte counts [92]. When new lesions developed, the patient benefited from a combination of PD-1 antibody and apatinib, achieving partial and complete responses in different lesions [92]. Subsequent maintenance therapy with a PD-1 antibody alone further improved outcomes, resulting in a durable clinical benefit over several years, highlighting the synergistic potential of radiotherapy, neoantigen-specific immunotherapy, and anti-angiogenic agents in achieving long-term disease control in advanced HCC [92]. A recent single-arm, open-label, phase 1/2 clinical trial with identifier NCT04251117 revealed that the personalized therapeutic cancer vaccine (PTCV) GNOS-PV02, administered alongside a plasmid-encoded interleukin-12 plus pembrolizumab, demonstrated promising clinical efficacy in patients with advanced HCC who had previously undergone treatment with a multityrosine kinase inhibitor [20]. An objective response rate of 30.6% was observed, including complete response in 8.3% of patients, with these outcomes related to the number of neoantigens incorporated into the vaccine [20]. Robust neoantigen-specific T cell responses were detected in 86.4% of patients evaluated, confirming the activation, proliferation, and cytotoxic activity of both CD4+ and CD8+ T cells [20]. Additional analyses revealed vaccine-driven expansion and tumor infiltration of T cell clones, which demonstrated reactivity to the encoded neoantigens, validating the vaccine’s ability to elicit a potent antitumor immune response [20]. Specifically, the study revealed significant T cell clonal expansion in the periphery, with a subset of these clones infiltrating the tumor microenvironment [20]. These expanded T cells exhibited a cytotoxic phenotype, characterized by the high expression of granzyme B and perforin, markers directly associated with tumor cell killing [20]. Functional assays further demonstrated the cytolytic activity of these T cells in response to vaccine-encoded neoantigens, strongly supporting the hypothesis of direct tumor targeting by the vaccine-induced immune response [20]. Additionally, engineered T cells expressing vaccine-specific TCRs displayed reactivity to patient-specific neoantigens, further validating the vaccine’s ability to directly target tumor cells [20]. The treatment was well-tolerated, with no severe toxicities or dose-limiting adverse events reported, suggesting that PTCV combined with anti-PD-1 therapy can enhance clinical responses in advanced HCC [20]. A pilot phase I clinical trial assessed the safety and efficacy of a personalized neoantigen-pulsed autologous DC vaccine (Neo-DCVac-02) as an adjuvant treatment for high-risk HCC following radical surgery [93]. The treatment was well-tolerated, with no grade 2 or higher treatment-related adverse events. Among the thirteen patients evaluated, five exhibited enhanced antigen-specific T cell activity targeting more than 50% of the neoantigen peptides, with durable cellular immune responses persisting for up to one year [93]. Flow cytometric analysis revealed increased T cell activation and co-stimulation, indicating successful immune priming [93]. At a median follow-up of 29.7 months, the median relapse-free survival (RFS) was not reached, with one- and two-year RFS rates of 84.6% and 60%, respectively, and all patients remaining alive [93]. Notably, patients with a robust immune response achieved significantly longer RFS compared to those with weaker responses [93]. These findings highlight the safety and potential efficacy of Neo-DCVac-02 in inducing sustained antigen-specific immunity and delaying HCC recurrence [93]. A Chinese clinical trial with registration number ChiCTR1900020990 evaluated a personalized neoantigen vaccine as a prophylactic strategy to prevent postoperative recurrence in HCC patients at high risk of relapse [94]. Among 10 enrolled patients, the vaccine was well-tolerated with no significant adverse events reported [94]. At the end of the clinical trial, eight patients experienced disease recurrence, while two remained relapse-free [94]. The median RFS from the first vaccination was 7.4 months. Notably, five of the seven patients who completed the full vaccination schedule exhibited neoantigen-induced T cell responses and achieved significantly longer RFS compared to those without responsive neoantigens or those receiving only partial vaccinations [94]. Additionally, tracking neoantigen mutations in circulating tumor DNA (ctDNA) provided a dynamic and personalized measure of treatment efficacy and disease progression [94]. These findings demonstrate that personalized neoantigen vaccines are a safe, effective, and feasible approach to reducing postoperative HCC recurrence, while ctDNA monitoring offers valuable insights for improving individualized treatments [94].

## 5. Neoantigen-Based Vaccines and Immune Checkpoint Inhibitors in Hepatocellular Carcinoma

Neoantigen-based therapies, such as personalized vaccines, address key limitations of existing treatments for HCC, including ICIs, by enhancing tumor immunogenicity and overcoming immune evasion mechanisms. Neoantigens are highly specific to cancer cells and absent from healthy tissues, ensuring targeted immune activation [44]. While ICIs alone show limited efficacy due to low T cell infiltration and the immunosuppressive tumor microenvironment in HCC, neoantigen-based vaccines directly activate cytotoxic T cells by presenting tumor-specific antigens [44]. This leads to improved T cell infiltration, epitope spreading, and sustained immune responses. Moreover, when combined with ICIs, neoantigen vaccines can synergize to enhance antitumor immunity by reversing T cell exhaustion and amplifying effector functions [95]. Neoantigen vaccines stimulate robust T cell responses by presenting tumor-specific antigens, while ICIs inhibit immune checkpoint pathways (e.g., PD-1/PD-L1), preventing T cell exhaustion and enhancing antitumor activity. This synergy can improve T cell infiltration into the tumor microenvironment and sustain cytotoxic responses. Emerging data have underscored the synergistic potential of combining personalized neoantigen vaccines with ICIs. The rationale is that T cell stimulation through neoantigen-based vaccines and the combined use of ICIs which can inhibit the downregulation of T cells by tumor cells could possibly have a synergistic antitumor effect. Despite HCC’s generally low TMB and a low frequency of presented neoantigens, clinical trials have demonstrated encouraging outcomes when combining these therapeutic modalities. Preclinical HCC models have demonstrated that personalized neoantigen vaccines combined with PD-1 inhibitors significantly enhanced CD8+ tissue-resident memory T cell infiltration, reduced regulatory T cells and monocytic myeloid-derived suppressor cells, and led to increased tumor regression overall [96]. Additionally, in the phase I/II trial (NCT04251117) which evaluated the effect of DNA vaccine GNOS-PV02 in combination with pembrolizumab and plasmid-encoded IL-12, as mentioned above, a 30.6% response rate was achieved in patients with advanced HCC [20]. Additionally, neoantigen-specific robust T cell responses were identified in most HCC patients, while cellular profiling confirmed the activation, proliferation, and cytolytic activity of vaccine-specific CD4+ and CD8+ effector T cells [94]. Additionally, other ongoing clinical trials are investigating neoantigen vaccines in combination with ICIs for HCC. These include antibodies targeting the PD-1/PD-L1 axis (NCT04912765, NCT05761717, and NCT05269381) or combinations targeting both PD-1/PD-L1 and CTLA-4 (NCT04248569).

These findings suggest that combining neoantigen vaccines with ICIs may enhance tumor-specific immune responses by increasing T cell infiltration and countering the immunosuppressive tumor microenvironment. However, no direct comparative studies have been conducted to assess whether these combination therapies are superior to ICIs alone, which currently remain a standard treatment for advanced HCC. Given the limitations of available data, the superiority of combination therapies over ICIs alone cannot be conclusively determined. Future developments in neoantigen vaccine strategies should prioritize addressing several key challenges. First, improved patient stratification is needed to identify predictive biomarkers, such as high affinity neoantigens or specific pathway aberrations, that can guide personalized treatment approaches. Second, the integration of neoantigen vaccines with other therapeutic modalities, such as locoregional therapies, could further enhance their efficacy by modulating the tumor immune microenvironment. Third, randomized controlled trials comparing combination therapies to ICIs alone are essential to establish their relative benefits. Finally, overcoming tumor heterogeneity and variability in neoantigen expression will be critical for optimizing the therapeutic potential of neoantigen-based approaches.

## 6. Conclusions

Advancements in techniques such as neoantigen discovery, tumor genome sequencing, and the development of neoantigen-based immunotherapies have been pivotal in creating personalized cancer vaccines and adoptive cell therapies. The use of next-generation sequencing allows for the rapid and cost-effective identification of tumor-specific mutations. Additionally, algorithms designed to predict MHC-binding epitopes have made it possible to identify potentially immunogenic neoepitopes [97]. These technological innovations have enabled the creation of highly personalized cancer therapies tailored to target the unique tumor profile of each patient.

Neoantigen-based vaccines represent a transformative strategy for HCC management by utilizing tumor-specific somatic mutations to produce robust immune responses. These vaccines, combined with advancements in genomic sequencing and immune profiling, have demonstrated the potential to overcome the immunosuppressive tumor microenvironment in HCC. Preclinical experimental studies along with early-phase clinical trials highlight their ability to activate T cells, suppress T regulatory cells, and promote antitumor immunity. Although the clinical translation of these therapies seems challenging, including the optimal delivery methods, cost-effectiveness, and addressing of tumor heterogeneity, their integration with ICIs offers a promising avenue for achieving sustained responses and improving survival outcomes not only in advanced HCC patients, but also as an adjuvant treatment in patients with high recurrence risk after liver resection. Furthermore, prospective trials are eagerly needed in order to assess their effectiveness in different HCC histological phenotypes, especially in immune-excluded HCCs, where the intratumoral presence of regulatory T cells can inhibit the accumulation of activated cytotoxic T cells and evade the immune responses to neoantigen-based vaccines. It is possible that the combination of these promising agents with treatments that can alter the tumor immune microenvironment, such as locoregional therapies, could further enhance immune response and therapeutic outcomes. Continued research and clinical trials will be critical in improving these personalized therapies and uncovering their full potential in precision medicine.

## Figures and Tables

**Figure 1 cancers-17-00376-f001:**
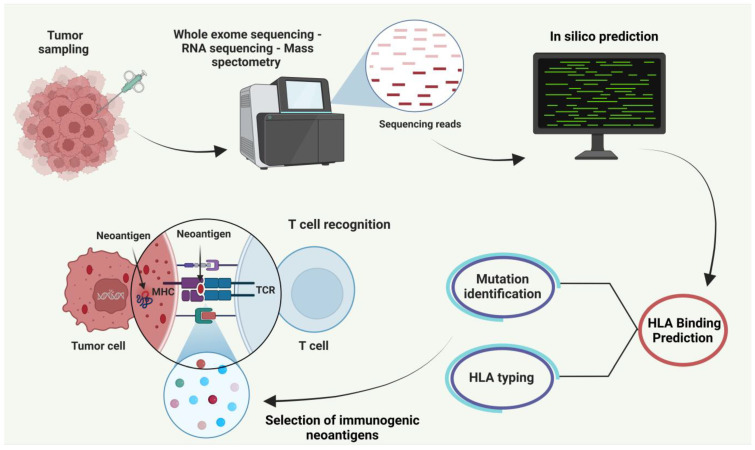
Workflow for neoantigen identification. Figure 1 illustrates the process for identifying and utilizing neoantigens in cancer immunotherapy. Tumor samples are collected and subjected to sequencing techniques such as whole-exome sequencing, RNA sequencing, or mass spectrometry to identify mutations. The sequencing data is processed using in silico prediction tools to determine tumor-specific mutations and their corresponding neoantigens. HLA binding prediction is then used to assess the affinity of these neoantigens for major histocompatibility complex (MHC) molecules. Following mutation identification and HLA typing, the most immunogenic neoantigens are selected based on their ability to be presented effectively by MHC molecules. These neoantigens are presented by tumor cells or antigen-presenting cells (APCs) to T cells, leading to T cell recognition and activation. This activation forms the basis for immune responses against tumor cells, driving the development of personalized cancer vaccines. Created with BioRender.com (accessed on 5 December 2024). Abbreviations: TCR, T cell receptor; MHC-I, major histocompatibility complex I; HLA, human leukocyte antigen.

**Figure 2 cancers-17-00376-f002:**
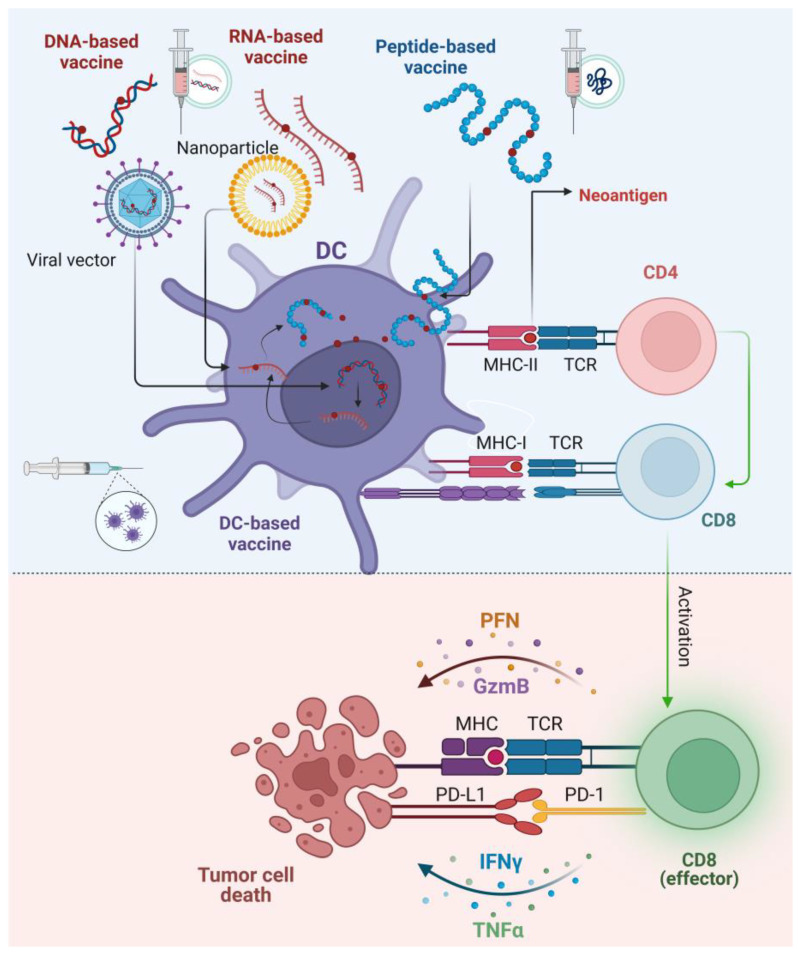
Neoantigen-based vaccines: mechanisms and immune modulation in cancer therapy. Figure 2 illustrates the mechanisms of neoantigen-based vaccines and their role in cancer immunotherapy. Tumor-specific mutations result in the generation of neoantigens, which are delivered through various vaccine platforms, including DNA-based, RNA-based, peptide-based, and dendritic cell (DC)-based vaccines. These vaccines enable dendritic cells to process and present neoantigens via MHC-II to CD4+ helper T cells and MHC-I to CD8+ cytotoxic T cells. CD4+ T cells support the activation and expansion of CD8+ T cells, which then target and kill tumor cells by releasing perforin (PFN) and granzyme B (GzmB). These molecules induce apoptosis in tumor cells. Additionally, cytokines such as interferon-gamma (IFNγ) and tumor necrosis factor-alpha (TNFα) amplify the immune response, enhancing tumor cell destruction. The figure also highlights the immune checkpoint interaction between PD-1 on CD8+ T cells and PD-L1 on tumor cells, which can inhibit T cell activity. The same interactions are observed with CTLA4 and CD80/86 between T cells and tumor cells. Immune checkpoint inhibitors, such as anti-PD-1, anti-PD-L1, and anti-CTLA4 therapies, can block this interaction, enhancing T cell function and improving vaccine efficacy. Created with BioRender.com (accessed on 13 January 2024). Abbreviations: DC, dendritic cell; CD4, T helper cells; CD8, T cytotoxic cells; TCR, T cell receptor; MHC-I, major histocompatibility complex I; MHC-II, major histocompatibility complex II; PFN, perforin; GzmB, granzyme B; IFNγ, interferon-gamma; TNFα, tumor necrosis factor-alpha.

**Table 1 cancers-17-00376-t001:** Mechanisms of neoantigen generation.

Mechanism	Description	Impact on Neoantigen Immunogenicity
Single Nucleotide Variants (SNVs)	Point mutations that result in altered amino acids, generating tumor-specific epitopes	Moderate to high—depends on mutation type and MHC binding
Insertions/Deletions (Indels)	Frameshift or in-frame changes that create novel peptide sequences absent in normal tissues	High—often generates unique peptides with strong MHC affinity
Translocations	Rearrangements between chromosomes that can create fusion proteins or new antigenic sequences	Variable—depends on the fusion product’s presentation by MHC molecules

**Table 2 cancers-17-00376-t002:** Overview of ongoing clinical trials investigating neoantigen-based vaccines for hepatocellular carcinoma (HCC).

Clinical Trial ID	Clinical Trial Phase	Type of HCC	Vaccine Type	Vaccine Name	Clinical Trial Status	Study Initiation Date
NCT05981066	NA	Advanced (HCC) (Relapsed/refactory)	mRNA Vaccine	ABOR2014 (IPM511)	Recruiting	10 July 2023
NCT03674073	Phase I	HKLC stage IIa HCC	DC Vaccine & microwave ablation	NA	Unknown status	15 October 2018
NCT04912765	Phase II	Resectable HCC (group A) or CRLM (group B)	DC Vaccine & Nivolumab	NA	Recruiting	15 April 2021
NCT05105815	Early Phase I	High Risk of Recurrence After Surgical Resection of Primary HCC	NeoAg/aeTSA CTL	IPM001	Not yet recruiting	31 December 2021
NCT05536427	Phase I	Advanced HCC	NeoAg/aeTSA CTL	IPM001	Unknown status	1 October 2022
NCT05761717	NA	Post Operative HCC	mRNA Vaccine & Sintilimab	NA	Not yet recruiting	20 April 2023
NCT04248569	Phase I	Fibrolamellar HCC	Peptide Vaccine & Nivolumab & Ipilimumab	DNAJB1-PRKACA Fusion Kinase	Recruiting	20 April 2020
NCT05269381	Phase I/II	Advanced Solid Tumors (PNeoVCA) (including HCC)	Peptide Vaccine & Pembrolizumab	NA	Recruiting	31 March 2022
NCT04147078	Phase I	Postoperative Cancer (including HCC)	DC Vaccine	NA	Recruiting	1 June 2019
NCT04251117	Phase I/IIa	Advanced HCC	DNA vaccine & plasmid encoded IL-12 (INO-9012) & pembrolizumab (MK-3475)	GNOS-PV02 INO-9012	Completed	1 March 2020
NCT03199807	Phase IB/II	Advanced HCC (LCRAI-1)	NRT & Radiotherapy	NA	Unknown status	20 July 2017
ChiCTR1900020990	NA	Primary hepatocellular carcinoma	Peptide Vaccine	NA	Ongoing Recruiting	24 January 2019

This information is available on https://clinicaltrials.gov/ and on https://www.chictr.org.cn/indexEN.html (accessed on 3 December 2024). Abbreviations: CRLM, Colorectal liver metastasis; NeoAg/aeTSA CTL, neoantigen/altered epitope tumor-specific antigen cytotoxic T lymphocytes; NRT, New Antigen Reactive Immune Cells; LCRAI-1, Liver Imaging Reporting and Data System 1A.

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
