# Peer review of "Personalized Immunity: Neoantigen-Based Vaccines Revolutionizing Hepatocellular Carcinoma Treatment"

_cancers, 2025, doi:10.3390/cancers17030376_

Round 1
Reviewer 1 Report
Comments and Suggestions for Authors
The authors reviewed the promising potential of personalized neoantigen vaccines in the treatment of liver cancer, emphasizing their role in enhancing immune responses to tumor-specific targets. The study explored the development and integration of these vaccines with other therapies, such as immune checkpoint inhibitors, to improve patient outcomes. While early clinical trials show promising results, the authors acknowledge challenges in tumor variability and vaccine production. Nonetheless, the findings highlight the importance of advancing personalized therapeutic strategies for liver cancer, offering hope for improved survival rates and long-term disease control. I have few suggestions for improvement.
1) Can the authors clarify how neoantigen-based therapies directly address the limitations of existing HCC treatments like ICIs?
2) Could you add specific examples or data demonstrating the higher immunogenicity of SNV-derived or indel-derived neoantigens in HCC?
3) I suggest including a figure or table summarizing the mechanisms (SNVs, indels, translocations) for easier understanding?
4) Can the authors elaborate on the specific mechanisms by which tumor cells evade immune detection, beyond immune selection and immunoediting?
5) Can the authors provide examples from recent clinical studies showing neoantigen-specific T-cell responses in HCC patients to strengthen the discussion?
6) Please provide more details on how neoantigen vaccines can be effectively combined with ICIs to enhance immune responses in HCC? It would be beneficial for the authors to discuss preclinical or clinical data supporting the combination of neoantigen vaccines with other immunotherapeutic strategies?
7) The article mentions conflicting results regarding the prognostic significance of TMB and neoantigen load in HCC. Could you elaborate on the specific factors that might explain the disparity in findings between the studies cited (e.g., different patient populations, sample sizes, or methodological differences)?
8) In the study by Liu et al., TMB was associated with improved OS, whereas a retrospective study indicated that higher TMB linked to poor prognosis. What are the potential biological mechanisms underlying these opposite findings? Could these differences be attributed to varying immune responses in patients at different disease stages?
9) The article highlights TP53-specific neoantigens as having a significant impact on OS and anti-tumor immunity. How do these TP53-specific neoantigens interact with the immune system to enhance cytotoxic lymphocyte infiltration? Are there any particular immune checkpoint markers that may influence this interaction?
10) While TMB and overall neoantigen load may have limited prognostic utility, high-affinity neoantigens (HANs) seem to correlate better with OS. Can you provide more insights into how HANs influence the immune system differently than other neoantigens? How can this be leveraged for targeted immunotherapy strategies?
11) The synergistic potential of combining neoantigen vaccines with immune checkpoint inhibitors (ICIs) has been discussed. Can you elaborate on the mechanistic rationale for this combination therapy in HCC? Specifically, how do ICIs like PD-1/PD-L1 blockers complement the effect of neoantigen vaccines in activating tumor-specific T-cell responses?
12) What evidence exists to suggest that the combination of neoantigen vaccines and ICIs leads to a durable clinical benefit in HCC patients? Are there any ongoing studies or preliminary data that indicate improved long-term survival rates with this combination treatment?
13) In the phase I clinical trial with GNOS-PV02 and pembrolizumab (NCT04251117), the study showed promising results with an objective response rate of 30.6%. What further improvements or modifications would you suggest to optimize these outcomes in future trials?
Author Response
Reviewer 1
The authors reviewed the promising potential of personalized neoantigen vaccines in the treatment of liver cancer, emphasizing their role in enhancing immune responses to tumor-specific targets. The study explored the development and integration of these vaccines with other therapies, such as immune checkpoint inhibitors, to improve patient outcomes. While early clinical trials show promising results, the authors acknowledge challenges in tumor variability and vaccine production. Nonetheless, the findings highlight the importance of advancing personalized therapeutic strategies for liver cancer, offering hope for improved survival rates and long-term disease control. I have few suggestions for improvement.
Response: Thank you for your thoughtful feedback and valuable suggestions. We appreciate your positive remarks on our review and have carefully addressed your suggestions in the revised manuscript to enhance its clarity and scientific depth.
Comment 1: Can the authors clarify how neoantigen-based therapies directly address the limitations of existing HCC treatments like ICIs?
Response to comment 1: We have added a paragraph in the beginning of the section 5 “Neoantigen-based Vaccines and Immune Checkpoint Inhibitors in Hepatocellular Car-cinoma”, in which we highlight how neoantigen-based therapies, such as personalized vaccines, address limitations of ICIs. It explains their mechanism of action, focusing on enhancing tumor immunogenicity and overcoming immune evasion (lines 480-489).
Comment 2: Could you add specific examples or data demonstrating the higher immunogenicity of SNV-derived or indel-derived neoantigens in HCC?.
Response to comment 2: We thank the reviewer for the suggestion. We have added data demonstrating the immunogenicity of SNV-derived neoantigens in HCC (lines 129-135). Regarding indels, we note that the manuscript already includes relevant data, specifically referencing the global proteogenomic analysis of MHC class I-associated peptides derived from non-canonical reading frames (lines 136-139).
Comment 3: I suggest including a figure or table summarizing the mechanisms (SNVs, indels, translocations) for easier understanding?
Response to comment 3: We have added a table presenting the mechanisms of neoantigen generation (Table 1) (line 118).
Comment 4: Can the authors elaborate on the specific mechanisms by which tumor cells evade immune detection, beyond immune selection and immunoediting?
Response to comment 4: In response, we have included additional mechanisms by which tumor cells evade immune detection, such as upregulation of immune checkpoint molecules (e.g., PD-L1), secretion of immunosuppressive cytokines (e.g., TGF-β, IL-10), recruitment of regulatory T cells (Tregs) and myeloid-derived suppressor cells (MDSCs), and downregulation of MHC molecules to impair antigen presentation (lines 168-173).
Comment 5: Can the authors provide examples from recent clinical studies showing neoantigen-specific T-cell responses in HCC patients to strengthen the discussion?
Response to comment 5: We thank the reviewer for the comment. In Section 4 of the manuscript “Neoantigen Vaccines in Hepatocellular Carcinoma”, we have already included examples from recent clinical studies, demonstrating neoantigen-specific T-cell responses in HCC patients. Specifically, the NCT04251117 trial reported robust neoantigen-specific T-cell responses in 86.4% of patients, with significant activation, proliferation, and cytolytic activity of vaccine-specific CD4+ and CD8+ T cells. Tumor-infiltrating T-cell clones exhibited cytotoxic phenotypes with high granzyme B and perforin expression (lines 440-448). The Neo-DCVac-02 trial showed enhanced antigen-specific T-cell activity in 5 out of 13 patients, with durable immune responses persisting for up to a year. This immune response correlated with prolonged relapse-free survival (lines 454-458). The ChiCTR1900020990 trial demonstrated that patients with neoantigen-induced T-cell responses had significantly longer relapse-free survival compared to non-responsive patients (lines 469-472). These findings underline the immunogenic potential of neoantigen-based therapies in eliciting effective T-cell responses in HCC patients.
Comment 6: Please provide more details on how neoantigen vaccines can be effectively combined with ICIs to enhance immune responses in HCC? It would be beneficial for the authors to discuss preclinical or clinical data supporting the combination of neoantigen vaccines with other immunotherapeutic strategies?
Response to comment 6: We have provided more details as suggested on how neoantigen vaccines can be effectively combined with ICIs to enhance immune responses in HCC (lines 487-493).
In Section 5 of the manuscript “Neoantigen-based Vaccines and Immune Checkpoint Inhibitors in Hepatocellular Carcinoma”, we have discussed the combination of neoantigen vaccines with ICIs and included clinical data supporting this approach. We have not included preclinical data to avoid extending the manuscript and going beyond the scope of this review. Our focus remains on clinical data, as numerous prior reviews have extensively described preclinical findings on this topic.
Comment 7: The article mentions conflicting results regarding the prognostic significance of TMB and neoantigen load in HCC. Could you elaborate on the specific factors that might explain the disparity in findings between the studies cited (e.g., different patient populations, sample sizes, or methodological differences)?
Response to comment 7: The disparity in findings regarding the prognostic significance of TMB and neoantigen load in HCC can be attributed to several factors. Differences in patient populations, including etiology (e.g., viral hepatitis vs. non-alcoholic fatty liver disease), tumor stages, and underlying liver conditions, may influence the outcomes. Additionally, variations in sample sizes, study designs, and methodologies, such as the use of different sequencing platforms or computational tools for neoantigen prediction, can contribute to inconsistent results. These data have been added to section 3.2 (lines 266-270).
Comment 8: In the study by Liu et al., TMB was associated with improved OS, whereas a retrospective study indicated that higher TMB linked to poor prognosis. What are the potential biological mechanisms underlying these opposite findings? Could these differences be attributed to varying immune responses in patients at different disease stages?
Response to comment 8: The opposite findings regarding the prognostic significance of TMB may be influenced by several potential biological mechanisms. In the study by Liu et al., higher TMB was associated with improved OS, potentially reflecting increased neoantigen load and enhanced immune activation in patients with earlier-stage disease. In contrast, the retrospective study linking higher TMB to poor prognosis may reflect the immunosuppressive tumor microenvironment in advanced HCC, where high TMB could promote tumor evolution, immune escape, and resistance to therapies. Differences in immune responses at various disease stages, such as greater T-cell infiltration and functional activity in earlier stages compared to immune exhaustion or suppression in advanced stages, could further explain these discrepancies (lines 270-276).
Comment 9: The article highlights TP53-specific neoantigens as having a significant impact on OS and anti-tumor immunity. How do these TP53-specific neoantigens interact with the immune system to enhance cytotoxic lymphocyte infiltration? Are there any particular immune checkpoint markers that may influence this interaction?
Response to comment 9: TP53-specific neoantigens are highly immunogenic due to their tumor-specific expression and the absence of tolerance mechanisms that typically limit responses to non-mutated antigens. These neoantigens enhance cytotoxic lymphocyte infiltration by being efficiently presented on MHC class I molecules, which are recognized by CD8+ T cells. This interaction stimulates the activation and proliferation of tumor-specific cytotoxic T lymphocytes (CTLs), contributing to robust anti-tumor immunity. Additionally, TP53-specific neoantigens have been associated with higher immune scores and increased infiltration of cytotoxic lymphocytes in HCC. Immune checkpoint markers, such as PD-1 and CTLA-4, may modulate this interaction. Inhibition of these checkpoints using ICIs can amplify the immune response by restoring T-cell functionality and preventing exhaustion, thereby enhancing the efficacy of TP53-specific neoantigen-driven immune responses. In response to the reviewer’s comment, we have added this information to the revised text (lines 284-291).
Comment 10: While TMB and overall neoantigen load may have limited prognostic utility, high-affinity neoantigens (HANs) seem to correlate better with OS. Can you provide more insights into how HANs influence the immune system differently than other neoantigens? How can this be leveraged for targeted immunotherapy strategies?
Response to comment 10: We have elaborated on the influence of HANs on the immune system in the context of neoantigen therapies and how HANs can be leveraged in immunotherapeutic strategies (lines 312-318 & 325-328).
Comment 11: The synergistic potential of combining neoantigen vaccines with immune checkpoint inhibitors (ICIs) has been discussed. Can you elaborate on the mechanistic rationale for this combination therapy in HCC? Specifically, how do ICIs like PD-1/PD-L1 blockers complement the effect of neoantigen vaccines in activating tumor-specific T-cell responses?
Response to comment 11: In the manuscript, we have added information how neoantigen vaccines synergize with ICIs by stimulating robust tumor-specific T-cell responses, while ICIs prevent T-cell exhaustion by inhibiting pathways like PD-1/PD-L1. This combination enhances T-cell infiltration into the tumor microenvironment and sustains cytotoxic responses, offering a strong mechanistic rationale for their use in HCC (lines 487-493).
Comment 12: What evidence exists to suggest that the combination of neoantigen vaccines and ICIs leads to a durable clinical benefit in HCC patients? Are there any ongoing studies or preliminary data that indicate improved long-term survival rates with this combination treatment?
Response to comment 12: Evidence supporting the combination of neoantigen vaccines and ICIs in HCC includes the NCT04251117 trial, which demonstrated a 30.6% objective response rate in patients with advanced HCC treated with a DNA neoantigen vaccine (GNOS-PV02) in combination with pembrolizumab and plasmid-encoded IL-12. The trial reported robust neoantigen-specific T-cell responses, tumor-infiltrating T-cell clonal expansion, and durable clinical benefits in some patients. Additionally, ongoing studies such as NCT04912765, NCT05761717, NCT05269381 and NCT04248569 are investigating the long-term efficacy of combining neoantigen vaccines with ICIs targeting PD-1/PD-L1 and CTLA-4 pathways, with the goal of improving survival rates. These studies highlight the potential of this combination therapy to enhance immune responses and achieve sustained clinical outcomes in HCC.
Comment 13: In the phase I clinical trial with GNOS-PV02 and pembrolizumab (NCT04251117), the study showed promising results with an objective response rate of 30.6%. What further improvements or modifications would you suggest to optimize these outcomes in future trials?
Response to comment 13: We have included a paragraph discussing the potential benefits and current limitations of combining neoantigen vaccines with ICIs (lines 513-528).
Reviewer 2 Report
Comments and Suggestions for Authors
Comments
This review has been well organized, and all paragraphs were summarized, generally considered appropriate, and easily understood. Authors described and discussed the wide contents, including the latest articles and clinical trials. The references were almost properly cited, although it was worrisome that there were many citations of reviews.
Major points
The combination therapies with ICI and neoantigen vaccine is considered as main approaches to HCC immunotherapy. However, HCC have generally low TMB and the frequency of presented neoatingens is about 0.5%, suggesting the low probability of effective neoantigens. Clinical trials showed the effect of combination therapies, but, we wondered whether the effect of combination therapies are superior compared with the current standard of treatment of ICI against HCC. Based on these findings, please discuss the issues, and the author's opinions for future developments of neoantigen vaccines should be added to the conclusion part.
Minor points
1. Introduction
l Authors described that HCC has low TMB and T-cell infiltration. But, although ref 14 showed low TMB in HCC, please add reference for low T-cell infiltration in HCC.
2. Tumor Neoantigen
2.1
l Polymorphisms (SNPs) should not be appropriate in this paragraph, because polymorphism means the variants whose exist frequency are more than 1% in germline cells. It is not a factor that produces somatic SNVs.
3. Neoantigen as Targets for Hepatocellular Carcinoma
l Although it is difficult that the immunogenicity of neoantigens are evaluated, in Figure 1, authors demonstrated the process called “selection of immunogenic neoantigen” after HLA binding prediction. What approaches are the process for “selection of immunogenic neoantigen”?
l The association between the abnormalities of multiple signaling pathways and prognosis in HCC has been described. Please add some explanations of the association between abnormalities in these pathways and the effect of neoantigen vaccines or the generation of neoantigen in HCC. Moreover, in the clinical trial of the neoantigen vaccine (+ICI), which is introduced in the next section, has there been a relationship between the abnormality of these pathways and the effect of the vaccine?
l The association of TP53-derived neoantigen with prognosis in HCC has been described. Is the combination with TP53-derived neoantigen vaccine with ICI effective? If there have been some trials, please refer to these reports.
l Reference 57 describes the association between HAN and CD39+CD8 TIL, but in this paper, HCC have high TMB. Is it due to differences in patient backgrounds? An explanation of HAN and the methodology for the prediction of HAN that corresponds to immunogenicity prediction should be added in this paragraph.
4. Neoantigen vaccine
l Reference 65 discussed the process of replacement of donor lymphocytes to host lymphocytes after liver transplantation with allograft. Reference 67 demonstrated that only 15% of patients with HCC had T cell population matching in peripheral and intratumoral. Please explain this discrepancy. The behavior of the donor lymphocytes in liver may be significantly differ from self-lymphocytes homeostasis.
l In Figure 2, what does "Ex vivo" mean? Does T cell priming by antigen-presenting cells occur after vaccination in vivo?
l The antitumor effect of IL12 on liver cancer has been reported as a activation of NK cells. In addition, autoimmune hepatitis is known as a typical side effect of ICI. As an effect of GNOS-PV02, an immune response to neoantigen has been observed in patients. Please discuss whether the neoantigen-specific immune response induced by vaccine directly acts on cancer.
l Paragraph 5 does not need to be discrived separately from Paragraph 4, since paragraph 5 only summarizes the conclusion of paragraph 4 again.
Author Response
Reviewer 2
This review has been well organized, and all paragraphs were summarized, generally considered appropriate, and easily understood. Authors described and discussed the wide contents, including the latest articles and clinical trials. The references were almost properly cited, although it was worrisome that there were many citations of reviews.
Response: Thank you for your kind feedback and thoughtful comments on our manuscript. We appreciate your positive remarks regarding its organization, clarity, and inclusion of the latest articles and clinical trials. In response to your concern about the use of review articles, we have carefully revised the manuscript and included several original studies to strengthen the references, addressing your comments and enhancing the scientific depth of our review.
Major Comments
Comment 1: The combination therapies with ICI and neoantigen vaccine is considered as main approaches to HCC immunotherapy. However, HCC have generally low TMB and the frequency of presented neoatingens is about 0.5%, suggesting the low probability of effective neoantigens. Clinical trials showed the effect of combination therapies, but, we wondered whether the effect of combination therapies are superior compared with the current standard of treatment of ICI against HCC. Based on these findings, please discuss the issues, and the author's opinions for future developments of neoantigen vaccines should be added to the conclusion part.
Response to comment 1: We thank the reviewer for the thoughtful comment regarding the need to discuss whether the effect of combination therapies involving neoantigen vaccines and ICIs is superior to ICIs alone, and to address future directions for neoantigen vaccine development. In response, we have incorporated this information into Section 5, Neoantigen-based Vaccines and Immune Checkpoint Inhibitors in Hepatocellular Carcinoma, directly before the conclusion (lines 513-528). This placement ensures that the discussion aligns naturally with the content of Section 5, where combination therapies are described in detail, allowing us to provide a more thorough and focused analysis. Including this topic in the conclusion section would not have allowed for the same level of detail and context, which we believe is essential for addressing the reviewer’s comment comprehensively.
Minor Comments
Comment 1: Introduction - Authors described that HCC has low TMB and T-cell infiltration. But, although ref 14 showed low TMB in HCC, please add reference for low T-cell infiltration in HCC.
Response to comment 1: We have added a reference supporting the observation of low T-cell infiltration in HCC to complement the existing data on low TMB (line 68).
Comment 2: Tumor Neoantigen – 2.1 Polymorphisms (SNPs) should not be appropriate in this paragraph, because polymorphism means the variants whose exist frequency are more than 1% in germline cells. It is not a factor that produces somatic SNVs.
Response to comment 2: We have revised as suggested.
Comment 3: Neoantigen as Targets for Hepatocellular Carcinoma - Although it is difficult that the immunogenicity of neoantigens are evaluated, in Figure 1, authors demonstrated the process called “selection of immunogenic neoantigen” after HLA binding prediction. What approaches are the process for “selection of immunogenic neoantigen”?
Response to comment 3: The selection of immunogenic neoantigens involves a combination of computational and experimental approaches to identify candidates capable of eliciting a strong immune response. Computational tools, such as HLA binding affinity prediction algorithms, are commonly used to estimate the binding affinity of neoantigen peptides to specific HLA molecules, with high-affinity binders prioritized as potential immunogenic candidates. Additionally, models can predict whether the neoantigen-HLA complex is likely to be recognized by T cell receptors (TCRs). Algorithms also assess the likelihood of neoantigens triggering an immune response based on various factors, such as binding strength and immunogenic potential. Finally, experimental validation using in vitro assays, such as T cell activation tests, or mass spectrometry is conducted to confirm the immunogenicity of the identified neoantigens.
Comment 4: Neoantigen as Targets for Hepatocellular Carcinoma - The association between the abnormalities of multiple signaling pathways and prognosis in HCC has been described. Please add some explanations of the association between abnormalities in these pathways and the effect of neoantigen vaccines or the generation of neoantigen in HCC. Moreover, in the clinical trial of the neoantigen vaccine (+ICI), which is introduced in the next section, has there been a relationship between the abnormality of these pathways and the effect of the vaccine?
Response to comment 4: Data have been added to explain the impact of pathway abnormalities on neoantigen generation and vaccine efficacy, as suggested (228-238). Regarding the potential relationship between pathway abnormalities and the effects of neoantigen vaccines in combination with ICIs, in the next section of the manuscript, we provide an overview of all existing clinical trials investigating neoantigen vaccines, including those combined with ICIs. However, given that the trials described involve multiple combinations and varying patient cohorts, the available data do not allow for a detailed analysis of specific pathway abnormalities and their direct correlation with vaccine efficacy.
Comment 5: Neoantigen as Targets for Hepatocellular Carcinoma - The association of TP53-derived neoantigen with prognosis in HCC has been described. Is the combination with TP53-derived neoantigen vaccine with ICI effective? If there have been some trials, please refer to these reports.
Response to comment 5: In response, we have expanded the discussion in the manuscript to address this point (292-310). Specifically, we acknowledge that while specific trials investigating the combination of TP53-derived neoantigen vaccines with ICIs in HCC are currently limited, there is growing evidence from preclinical and early-phase clinical studies in other cancers supporting the potential synergy of neoantigen vaccines and ICIs. Moreover, we have also included reference to the KEYNOTE-942 trial, a phase 2b study evaluating the combination of the mRNA-based neoantigen vaccine mRNA-4157 (V940) with pembrolizumab in patients with resected high-risk melanoma.
Comment 6: Neoantigen as Targets for Hepatocellular Carcinoma - Reference 57 describes the association between HAN and CD39+CD8 TIL, but in this paper, HCC have high TMB. Is it due to differences in patient backgrounds? An explanation of HAN and the methodology for the prediction of HAN that corresponds to immunogenicity prediction should be added in this paragraph.
Response to comment 6: Regarding your observation that the reference describes a high TMB in HCC, we would like to clarify that this is not the case. The reference explicitly states that TMB in HCC is low and does not correlate significantly with prognosis. As far as high affinity neoantigens (HANs), we have revised the paragraph to include a concise explanation of HANs and their prediction methodology (lines 312-322).
Comment 7: Neoantigen vaccine - Reference 65 discussed the process of replacement of donor lymphocytes to host lymphocytes after liver transplantation with allograft. Reference 67 demonstrated that only 15% of patients with HCC had T cell population matching in peripheral and intratumoral. Please explain this discrepancy. The behavior of the donor lymphocytes in liver may be significantly differ from self-lymphocytes homeostasis.
Response to comment 7: Thank you for your comment. To clarify, Reference 67 does not state that "only 15% of patients with HCC had T-cell populations matching in peripheral and intratumoral regions." Instead, it reports that "neoantigen-specific circulating CD8+ T cells (specifically targeting NY-ESO-1 and Glypican-3) are present in less than 15% of HCC cases." These findings are distinct: the former suggests a direct comparison or matching of T-cell populations between peripheral blood and intratumoral regions, while the latter refers solely to the presence of specific circulating CD8+ T cells in peripheral blood. This distinction indicates that the two statements refer to different aspects of T-cell biology and do not conflict.
Furthermore, the behavior of donor lymphocytes in liver allografts may significantly differ from the self-lymphocyte homeostasis observed in non-transplant settings. Donor lymphocytes, exposed to the immunological environment of the recipient, can undergo processes such as replacement, exhaustion, or adaptation distinct from the dynamics of self-lymphocytes. This divergence is particularly relevant in transplanted livers, where the immune microenvironment is modulated by factors such as immunosuppressive therapy, alloantigen exposure, and altered homeostatic mechanisms. These unique conditions may lead to discrepancies when comparing donor-derived and self-lymphocyte populations in terms of their peripheral and intratumoral matching behavior.
Comment 8: Neoantigen vaccine - In Figure 2, what does "Ex vivo" mean? Does T cell priming by antigen-presenting cells occur after vaccination in vivo?
Response to comment 8: We acknowledge that these labels may be misleading, as "ex vivo" applies specifically to dendritic cell (DC)-based vaccines, where antigen-presenting cells are loaded with neoantigens outside the body and subsequently reintroduced to the patient. In contrast, for RNA-, DNA-, and peptide-based vaccines, the entire process of T cell priming and activation occurs in vivo, as the vaccine directly stimulates the patient’s antigen-presenting cells within the body. To clarify this distinction, we have revised the figure by removing the generalized "in vivo" and "ex vivo" labels.
Comment 9: Neoantigen vaccine - The antitumor effect of IL12 on liver cancer has been reported as a activation of NK cells. In addition, autoimmune hepatitis is known as a typical side effect of ICI. As an effect of GNOS-PV02, an immune response to neoantigen has been observed in patients. Please discuss whether the neoantigen-specific immune response induced by vaccine directly acts on cancer.
Response to comment 9: We have discussed this, presenting the reasons why the neoantigen-specific immune response induced by the vaccine may directly act on cancer, as suggested by the reviewer (lines 440-448).
Comment 10: Neoantigen vaccine - Paragraph 5 does not need to be discrived separately from Paragraph 4, since paragraph 5 only summarizes the conclusion of paragraph 4 again.
Response to comment 10: We appreciate the reviewer’s suggestion to merge Paragraph 5 with Paragraph 4. However, we believe it is important to maintain these paragraphs as separate sections due to the distinct focus of each. Paragraph 4 provides a comprehensive overview of neoantigen vaccines in HCC, including their development, mechanisms of action, and general applications. In contrast, Paragraph 5 is dedicated to discussing the trials that specifically investigate the combination of neoantigen-based vaccines with ICIs, emphasizing their synergistic potential and unique therapeutic advantages. By keeping these sections separate, the manuscript ensures clarity and allows readers to easily distinguish between the broader applications of neoantigen vaccines and their integration with ICIs.
Round 2
Reviewer 1 Report
Comments and Suggestions for Authors
The authors addressed the concerns raised, the manuscript is suitable for publication